# Differential Vaginal Microbiota Profiling in Lactic-Acid-Producing Bacteria between Infertile Women with and without Chronic Endometritis

**DOI:** 10.3390/diagnostics12040878

**Published:** 2022-03-31

**Authors:** Suguru E. Tanaka, Yoshiyuki Sakuraba, Kotaro Kitaya, Tomomoto Ishikawa

**Affiliations:** 1Varinos Inc., DiverCity Tokyo Office Tower, 12F, 1-1-20 Aomi, Koto-ku, Tokyo 135-0064, Japan; stanaka@varinos.com; 2Reproduction Clinic Osaka, Grand Front Osaka Tower-A 15F, 4-20 Oofuka-cho, Kita-ku, Osaka 530-0011, Japan; kitaya@koto.kpu-m.ac.jp (K.K.); tishikawa@reposaka.jp (T.I.)

**Keywords:** chronic endometritis, dysbiosis, lactic acid bacteria, vaginal secretion microbiota

## Abstract

Purpose: Chronic endometritis (CE) is an infectious and inflammatory disorder associated with infertility of unknown etiology, repeated implantation failure, and recurrent pregnancy loss. In the current clinical practice, intrauterine interventions such as endometrial biopsy/histopathologic examinations and/or hysteroscopy are required for the diagnosis of CE. In this study, we analyzed the microbiota in vaginal secretions (VS) as a potential prediction tool for CE in infertile women. Methods: Using next-generation sequencing analysis, we compared the VS and endometrial fluid (EF) microbiota in infertile women with (*n* = 20) or without CE (*n* = 103). Results: The detection rate of *Streptococcus* and *Enterococcus* as well as the bacterial abundance of *Atopobium* and *Bifidobacterium* in the VS microbiota was significantly lower in the CE group than in the non-CE group. Meanwhile, the detection rate and bacterial abundance of *Lactobacillus* in the EF and VS microbiota were at similar levels between the two groups. Conclusion: These findings suggest that VS microbiota in infertile women with CE is characterized by the reduction in *Bifidobacterium* and lactic-acid-producing bacteria other than *Lactobacillus*. Our results hold promise for the prediction of CE, not by somewhat interventional intrauterine procedures, but by less invasive VS sampling. TRIAL REGISTRATION NUMBER: UMIN000029449 (registration date 6 October 2017).

## 1. Introduction

Chronic endometritis (CE) is a mucosal infectious/inflammatory disease with the unusual infiltration of CD138(+) endometrial stromal plasma cells (ESPC) [1]. CE also frequently presents with several unique hysteroscopic findings represented by endometrial micropolyposis and strawberry aspect [2,3]. Accumulating studies demonstrate that CE is identified in a substantial proportion of infertile patients with unknown etiology, repeated implantation failure (RIF) following in vitro fertilization–embryo transfer (IVF-ET) treatment, and recurrent pregnancy loss [3,4,5,6,7,8,9,10]. For example, women with a history of RIF/CE have a significantly lower implantation rate in the IVF-ET cycle than those with RIF but without CE (15% vs. 46%) [7]. In addition, the live birth rate in the subsequent pregnancy is very poor in women with a history of recurrent pregnancy loss/untreated CE (7%) [6]. Moreover, women of reproductive age who contracted CE were at 60% higher risk of future infertility compared with non-CE cohort [11].

The expression of multiple genes such as ovarian steroid receptors, adhesion molecules, cytokines, chemokines, and apoptosis are dysregulated in the secretory phase endometrium with CE, indicating the impaired endometrial receptivity in this pathology [12,13,14]. Furthermore, persistent CE is considered to lead to chronic deciduitis, a pathologic condition associated with preterm labor and neonatal periventricular leukomalacia/cerebral palsy [15,16,17,18,19,20,21,22]. Thus, CE is thought to have a negative impact on entire period of human reproduction.

The major cause of CE is intrauterine infection by pathogens including common bacteria, *Mycoplasma*, *Ureaplasma*, and *Mycobacterium Tuberculosis* [2,8], as antibiotic administration against these microorganisms has been proved to eradicate ESPC [6,7,8,9,10]. *Lactobacillus* is the bacterial genus comprising a majority in the vaginal cavity of healthy premenopausal women [23]. Lactic acid produced by vaginal *Lactobacilli* plays a crucial role in opposition against invading deleterious microorganisms via mucosal pH drop, active compound production, and host immune system elicitation [24]. In contrast, the microbial community (i.e., microbiota) in the human uterine cavity remains controversial. While some studies claimed the predominance of *Lactobacillus* over other bacterial genera in the endometrium and/or endometrial fluid (EF), others reported that the microbiota in the uterine cavity are more diverse than that in the vaginal cavity and are not always characterized as a *Lactobacillus*-dominant condition [25,26,27,28,29,30,31,32,33,34,35,36,37,38].

Our previous studies demonstrate that conventional endometrial tissue culture and PCR failed to identify microorganisms in more than half of cases of infertile women with CE [8]. Moreover, the microorganisms detected in the EF and/or endometrium are often inconsistent with those detected in the endocervix or vaginal secretions (VS) [39]. These findings implicate the limitation of traditional microbial sampling and examinations in the diagnosis of CE. The introduction of the microbiota analysis into clinical practice is being actively carried out in healthcare fields. Of note, the technique using next-generation sequencing allows for the identification and composition of non-culturable microorganisms. Although this new tool has a potential to disclose the relationship between CE and microbial communities in the uterine cavity, only a few studies attempted microbiota analysis in CE. In this study, we aimed to compare microbiota in paired VS and EF samples between infertile women with or without CE, which was diagnosed with histopathology/immunohistochemistry and hysteroscopy.

## 2. Materials and Methods

### 2.1. Subjects, Sample Collection, and Examinations

This was an interim analysis of an ongoing case–control study approved by the Ethical Committee of the Institutional Review Board (Approval Number 2017-02, 10 September 2017) and registered on a clinical trial registration website (UMIN000029449). 

Under given written informed consent, infertile women with and without a history of CE were enrolled in the study. They had undergone primary examinations including basal hormonal measurements, thyroid functions, hysterosalpingogram, thrombophilic, and immunological factors in advance. Hysteroscopy was performed using a 3.1 mm diameter flexible endoscope with continuous fluid flow (Pentax EPM300, Ricoh-imaging, Tokyo, Japan) in the proliferative phase (on days 6–12) of the menstrual cycle [5]. The endoscopic images were stored for analysis and discussion. The endometrial biopsy samples were obtained using a 3 mm curette (Atom-Medical, Tokyo, Japan), washed immediately, fixed overnight in 4% paraformaldehyde (in phosphate buffer, pH 7.3), embedded in paraffin, and cut into 4-μm-thickness sections onto slide glasses.

On days 6–8 after spontaneous luteinizing hormone surge or human chorionic gonadotropin trigger, or on day 5 following luteal support in the same menstrual cycle, the paired VS/EF samples were obtained, carefully avoiding contamination. In brief, following the perineum cleansing with benzalkonium chloride solution and intravaginal bivalve speculum insertion, VS were collected from all directions of the vaginal mucosa using a swab and snapped into stabilizing liquid (DNA Genotek, Ottawa, ON, Canada) for solubilization. After removing the residual mucous with sterilized cotton balls, the vaginal cavity and cervix/portio were cleaned with benzalkonium chloride. EF was directly aspirated using a pipette, avoiding touching the speculum and vaginal wall, and transferred into another collection tube [36]. 

### 2.2. Immunohistochemistry for CD138 and Diagnosis of CE

The endometrial sections were dewaxed in limonene (Falma, Tokyo, Japan) and rehydrated in a graded series of ethanol (in phosphate-buffered saline, pH 7.4). The sections were then subjected to microwave pretreatment in citrate buffer solution for 5 min to retrieve antigens and immersion in 3% H_2_O_2_ for 5 min to quench tissue endogenous peroxidase. After washing in phosphate-buffered saline, the sections were soaked in 10% fetal calf serum (SAFC Biosciences, Lenexa, KS, USA) for 10 min to suppress non-specific antibody binding and then incubated with mouse anti-CD138 monoclonal IgG antibody (B-A38; Nichirei, Tokyo, Japan) or control mouse IgG. After washing, the immunostaining was visualized with an LSAB kit (Dako, Kyoto, Japan). Following hematoxylin nuclear counterstaining, the sections were observed by an experienced gynecologic pathologist under a light microscope (400× magnification) and evaluated for ESPC (stromal CD138+ cells with nucleic heterochromatin pattern) in 20 or more high-power fields. The ESPDI (ESPC density index) was calculated as the sum of ESPC counts divided by the number of the high-power fields evaluated. Histopathologic CE was diagnosed as an ESPDI score of 0.25 or more [8]. Meanwhile, hysteroscopic CE was diagnosed according to the criteria proposed by the International Working Group for Standardization of Chronic Endometritis Diagnosis (Table 1) [40].

### 2.3. DNA Extraction and Sequencing

The VS/EF samples were pretreated with proteinase K (Beckman Coulter, Brea, CA, USA) containing 100 mg/mL lysozyme solution/RNaseA (Sigma-Aldrich, Darmstadt, Germany). Following the genomic DNA extraction, the double-stranded DNA concentration was quantified, and the variable region 4 of the bacterial 16S rRNA gene was amplified from DNA using a modified primer pair, 515f (5′-TCGTCGGCAGCGTCAGATGTGTATAAGAGACAGGTGYCAGCMGCCGCGGTAA-3′) and 806rB (5′-GTCTCGTGGGCTCGGAGATGTGTATAAGAGACAG-GGACTACNVGGGTWTCTAAT-3′), with Nextera XT (Illumina, San Diego, CA, USA) adapter overhang sequences [41]. PCR was performed using 25 ng of DNA, 200 μmol/L 4-deoxynucleotide triphosphates, 400 nmol/L of each primer, 2.5 U FastStart-HiFi-polymerase, 4% of 20 mg/mL bovine serum albumin, 0.5 mol/L betaine, and the appropriate buffer with MgCl_2_ (Sigma-Aldrich) under the following conditions; denaturation (94 °C, 2 min) followed by 30 cycles of denaturation (94 °C, 20 s), annealing (50 °C, 30 s), extension (72 °C, 60 s), and final extension (72 °C, 5 min). Following purification using Agencourt AMPure XP (Beckman Coulter), the PCR products were multiplexed with a dual-index approach with a Nextera XT Index kit v2 and indexed using HiFi-HotStart-ReadyMix (Kapa Biosystems, Boston, MA, USA). The final library was sequenced at 2 × 200-bp using a MiSeq Reagent Kit v3 (Illumina). The ZymoBIOMICS Microbial Community Standard (a mixture of *Pseudomonas*, *Escherichia*, *Salmonella*, *Lactobacillus*, *Enterococcus*, *Listeria*, *Bacillus*, and yeast *Saccharomyces* and *Cryptococcus*, Zymo Research, Orange, CA, USA) was used as a positive control, whereas UltraPure™ DNase/RNase-Free-distilled water (ThermoFisher Scientific, Waltham, MA, USA) was used as a blank control. Using EA-Utils fastq-join [42], the median 291-base pair merged sequence length was obtained. The sequence quality control was performed using USEARCH v10.0.240 [43] to remove PhiX reads, truncate primer-binding sequences, and discard sequences with <100 bp length and sequence quality <Q20. Quantitative Insights Into Microbial Ecology 1.9.1 [44] was used with the default parameters for quality filtering, chimera check, and sequence clustering into operational taxonomic units (OTUs) with the open-reference picking strategy/UCLUST method based on 97% sequence identity [45]. Taxonomy was assigned to each OTU using a Ribosomal Database Project Classifier [46] with a 0.50 confidence threshold against the Greengenes database version 13_8 [47]. Fifteen bacterial taxa (*Acidovorax*, *Acinetobacter*, *Chryseobacterium*, *Citrobacter*, *Elizabethkingia*, *Escherichia*, *Flavobacterium*, *Janthinobacterium*, *Leptothrix*, *Methylobacterium*, *Pseudomonas*, *Rhodococcus*, *Sphingomonas*, *Stenotrophomonas*, and *Yersinia*), the known contaminants found in a blank control [48,49,50], were excluded from ES samples.

### 2.4. Statistics

The Shannon index and Chao1 richness at the 3000 sequences were adopted as α-diversity indices. Permutational multivariate analysis of variance was utilized for β-diversity indices. The normal distribution of the data sets was assessed using the Shapiro–Wilk test. Fisher’s exact test and Wilcoxon rank-sum test were performed to compare the detection rate and bacterial abundance between the groups, respectively. The adjusted Welch’s *t* test was conducted using arcsine transformed values of each bacteria percentage for normality assumption. All *p* values were calculated using two-sided tests with Excel Statistics software (SSRI, Tokyo, Japan) and regarded as statistically significant when the values were less than 0.05.

## 3. Results

### 3.1. Characteristics of Patients

A total of 130 infertile women were enrolled in the study between October 2017 and August 2019. Seven patients were excluded due to the insufficiency of either VS or EF, resulting in 123 patients being included in the analysis. While 83 patients had a history of RIF, 40 patients were infertile women undergoing their first IVF-ET attempt who volunteered to learn about their own VS and EF microbiota. All the patients were Japanese. There were no cigarette smokers and obese women (body mass index > 30). Their demographics, infertility etiology, and history are summarized in Table 2.

Histopathologic CE was diagnosed in a total of 20/123 (16.2%) patients according to the presence of CD138(+) ESPC. In these 20 cases of histopathologic CE, a total of 8 patients had some hysteroscopic abnormal findings associated with CE (Table 1). While there were no differences in demographics including age, body mass index, gravidity, and parity between the two groups, the proportion of the women with etiology of unexplained infertility was significantly higher in the CE group than in the non-CE group (*p* = 0.042, odds ratio 2.95, 95% confidence interval 1.06−8.20). The short gonadotropin-releasing hormone agonist protocol was used in more cycles in the CE group than in the non-CE group, but the difference between the two groups did not reach a significant level (*p* = 0.0502). 

The paired VS/EF samples were obtained from the CE group (*n* = 20; 7 in the natural cycle, 2 in the oocyte pick up cycle, and 11 in the hormone replacement cycle) and the non-CE group (*n* = 103; 32 in the natural cycle, 27 in the oocyte pick up cycle, and 44 in the hormone replacement cycle) and were subjected to sequencing. There were no significant differences in the proportion of the natural cycle, oocyte pick up cycle, and hormone replacement cycle (*p* > 0.15)

### 3.2. Sequencing Results of VS and EF Samples

Sequencing was successful in all VS/EF samples both in the CE and non-CE group. The raw sequence reads per sample were significantly (*p* < 2.66 × 10^−11^) lower in EF (mean 110,965, range 5969–474,550 reads) than those in VS (mean 166,965, range 60,484–2,167,046 reads). Following the quality filtering, the sequence reads per sample were yet (*p* < 3.64 × 10^−11^) lower in EF (mean 66,899, range 3302–341,036 reads) than those in VS (mean 124,110, range 27,067–1,280,317 reads). Finally, the microbiota in EF obtained was a mean 56,133 OTUs assigned sequences per sample (mean, range 2991–340,098 reads), whereas the microbiota in VS obtained was a mean 123,891 OTUs assigned sequences per sample (range 27,041–1,273,732 reads). Thus, the number of OTUs assigned sequences in the EF microbiota was 2.2-fold lower on average than those in the VS microbiota.

### 3.3. Diversity Comparison of VS and EF Microbiota between Infertile Women with and without CE

Rarefaction analysis demonstrated that the Shannon index was highly stable above 1000 sequences, indicating that enough sequencing was conducted to analyze the diversity of both the VS and EF microbiota (Figure 1A,B). The mean ± standard error (SE) Shannon index in the EF microbiota was 2.01 ± 0.19 in the CE group and 2.21 ± 0.09 in the non-CE group at 3000 reads (Figure 1C). There were no differences in the Shannon index in the EF microbiota between the two groups (*p* = 0.38). The richness of bacterial community in the EF microbiota measured using Chao1 richness was also at a similar level (*p* = 0.30) between the CE (990 ± 353) and non-CE groups (715 ± 94) (Figure 1D) as well as the phylogenetic diversity whole tree (*p* = 0.29, 3.71 ± 0.28 in the CE group and 4.08 ± 0.14 in the non-CE group, Figure 1E). The β-diversity indices indicated that there were no significant differences in the EF microbiota between the CE and non-CE group (*p* = 0.55, Figure 1I).

In addition, the Shannon index in the VS microbiota was similar (*p* = 0.66) between the CE group (1.74 ± 0.18) and the non-CE group (1.84 ± 0.09) (Figure 1F), along with Chao1 richness (*p* = 0.89, 897 ± 280 in the CE group and 855 ± 127 in the non-CE group, Figure 1G) and phylogenetic diversity whole tree (*p* = 0.21, 2.45 ± 0.22 in the CE group and 2.84 ± 0.13 in the non-CE group, Figure 1H). The β-diversity indices in the VS microbiota were also at a similar level between the two groups (*p* = 0.46) (Figure 1J).

### 3.4. Comparison of Bacterial Genera in EF Microbiota between Infertile Women with and without CE

The heatmap diagrams representing bacterial genera detected (with a total rate 0.1% or more) in VS and EF in the CE and non-CE groups are shown in Figure 2.

*Rhodanobacter* was detected in 15.0% (3/20) of the EF microbiota in the CE group, whereas it was detectable in 1.9% (2/103) of the EF microbiota in the non-CE group (Table 3). The detection rate (*p* = 0.030, by Fisher’s exact test, OR 8.91, 95% CI 1.39–57.3) and the bacterial abundance (*p* = 0.0062, by Wilcoxon rank-sum test) of *Rhodanobacter* in the EF microbiota of the CE group was significantly higher than in that of the non-CE group. 

In contrast, by the adjusted Welch’s *t* test, the bacterial abundance of *Atopobium* (*p* = 0.0022), *Bifidobacterium* (*p* = 0.0054), *Aeromonadaceae* (*p* = 0.014), *Vibrio* (*p* = 0.037), and *Clostridiales* (*p* = 0.048) was significantly lower in the EF microbiota in the CE group than in the non-CE group, although there were no differences in their detection rate (*p* > 0.20, by Fisher’s exact test) (Table 3).

*Lactobacillus* was detected in 90.0% (18/20) of the EF microbiota in the CE group, whereas it was detected in 94.2% (97/103) of the EF microbiota in the non-CE group. The detection rate (*p* = 0.62, by Fisher’s exact test, OR 0.56, 95% CI 0.09–6.10) and bacterial abundance (*p* = 0.29, by Wilcoxon rank-sum test) of *Lactobacillus* in the EF microbiota was at a similar level between the two groups. *Burkholderia,* a proteobacteria that was detected in a quarter of EF samples in infertile women with a history of RIF, but not in those undergoing their first IVF-ET attempt, was detectable in the EF microbiota at a similar level (*p* = 0.64, OR 1.52, 95% CI 0.14–8.89) between the CE group (10%, 2/20) and non-CE group (8%, 7/103). 

### 3.5. Comparison of Bacterial Genera in VS Microbiota between Infertile Patients with and without CE

*Streptococcus* was detected in 5.0% (1/20) of the VS microbiota in the CE group, whereas it was detectable in 38.8% (40/103) of the VS microbiota in the non-CE group (Table 3). The detection rate (*p* = 0.0033, by Fisher’s exact test, OR 0.08, 95% CI 0.004–0.52) and bacterial abundance (*p* = 0.0073, by Wilcoxon rank-sum test) of *Streptococcus* in the VS microbiota was significantly lower in the CE group than in the non-CE group.

In addition, *Enterococcus* was not detected in any (0/20) of the VS microbiota in the CE group, whereas it was detectable in 17.5% (18/103) of the VS microbiota in the non-CE group. The detection rate (*p* = 0.042, by Fisher’s exact test) and bacterial abundance (*p* = 0.045, by Wilcoxon rank-sum test) of *Enterococcus* in the VS microbiota was significantly lower in the CE group than in the non-CE group (Table 3). 

By the adjusted Welch’s *t* test, the bacterial abundance of *Atopobium* (*p* = 0.0029), *Bifidobacterium* (p = 0.0044), *Nesterenkonia* (*p* = 0.019), *Anaerococcus* (*p* = 0.024), and *Staphylococcus* (*p* = 0.033) was significantly lower in the VS microbiota in the CE group than in the non-CE group (Table 3), although there were no differences in their detection rate (*p* > 0.071).

Meanwhile, *Lactobacillus* was detected in 84.5% (87/103) of the VS microbiota in the CE group, whereas it was detected in 90.0% (18/20) of the VS microbiota in the non-CE group. The detection rate (*p* = 0.73, by Fisher’s exact test, OR 1.65, 95% CI 0.37–10.86) and the bacterial abundance (*p* = 0.19, by Wilcoxon rank-sum test) of *Lactobacillus* in the VS microbiota was at a similar level between the two groups.

### 3.6. Comparison of Lactic-Acid-Producing Bacteria in Microbiota between Infertile Patients with and without CE

We focused on the combination of four lactic-acid-producing bacteria, *Streptococcus**, Enterococcus*, *Bifidobacterium**,* and *Atopobium*, of which the detection rate and/or bacterial abundance in the VS microbiota significantly differed between the CE group and non-CE group. The specificity for the diagnostic value of CE was 0.96 when the total abundance of the combination of these lactic-acid-producing bacterial genera in the VS microbiota was set at 12.7% (*p* = 0.038 by Fisher’s exact test, sensitivity 0.29, accuracy 0.41). For the prediction of CE, the minimal *p* value (*p* = 0.0021, specificity 0.91, sensitivity 0.47, accuracy 0.55) was obtained when the combined proportion of *Streptococcus*, *Enterococcus*, *Bifidobacterium*, and *Atopobium* in the VS microbiota was set at 0.1%.

## 4. Discussion

Using endometrial tissue/EF collected on day 7 after luteinizing hormone surge in the identical menstrual cycle, Liu et al. [51] analyzed the relationship between CE and EF microbiota in infertile women of southeastern Chinese origin. While the average proportion of *Lactobacillus* in the EF was 80.7% in 118 non-CE infertile women, its proportion was only 1.89% in 12 CE women. The EF in CE patients was characterized by the richness of 18 taxa including *Dialister*, *Bifidobacterium*, *Prevotella*, *Gardnerella*, *Anaerococcus*, *Sphingomonadaceae*, *Corynebacterium*, *Micrococcus*, *Tepidimonas*, *Peptoniphilus*, *Howardella*, *Varibaculum*, *Kocuria*, *Psychrobacter*, *Peptoniphilus*, and *Luteimonas*, along with *Lactobacillus iners*. The presence of *Anaerococcus* and *Gardnerella* in the EF microbiota in CE patients were negatively correlated in relative abundance with *Lactobacillus*. Meanwhile, *Lactobacillus crispatus* was less abundant in CE than in non-CE women. One limitation of their study was the comparison between women with and without CE based solely on histopathologic criteria, but not on other methods such as microbial culture or hysteroscopy. In addition, vaginal microbiota was not examined as a reference to exclude the potential contamination from the lower genital tract.

In this study, we analyzed the paired VS and EF microbiota in infertile CE and non-CE patients from the Japanese population. We centered on histopathology/immunohistochemistry for CD138 for the diagnosis of CE, but also performed hysteroscopic diagnosis. As both diagnostic methods have advantages and disadvantages, we could not currently conclude the superiority of one method over another. We collected endometrial tissue in the proliferative phase (on days 6–12) of the menstrual cycle for the diagnosis of CE, as accumulating studies demonstrated that the diagnostic rate of CE is better in the preovulatory period than the postovulatory/implantation period [1,52,53,54]. Sequencing was successful in all paired samples. Unlike our recent report that the α- and β-diversity in the EF microbiota differ between infertile women with and without a history of RIF [36], these diversity indices were similar between the CE and non-CE group. The detection rate of *Burkholderia*, a genus of proteobacteria frequently found in the EF microbiota in infertile women with a history of RIF [36], did not differ between the two groups. Such inter-study variabilities are likely to result from the patient variance in the current study, where only 64% of the patients had a history of RIF. 

Intriguingly, in striking contrast to the study by Liu et al. [51], our findings demonstrated that CE is not associated with the non-*Lactobacillus* dominant condition in the EF microbiota, as its bacterial abundance in the VS and EF microbiota were at a similar level between the CE and non-CE group. Furthermore, on the contrary to their findings that *Bifidobacterium* was relatively rich in the EF microbiota in CE, our results showed the bacterial abundance of *Bifidobacterium* was significantly lower in the EF microbiota in the CE group than in the non-CE group, although there were no differences in its detection rate. A recent study using the samples obtained via the trans-peritoneo-myometrial route demonstrated the endometrium (and cervix) microbiota in the mid-secretory phase are dominated by *Acinetobacter*, *Pseudomonas*, *Cloacibacterium*, and *Comamonadaceae*, whereas *Lactobacillus* species (*L. iners* and *L. crispatus*) are rarely detectable in the endometrium [37]. Moreover, others using the same trans-peritoneo-myometrial sampling route showed six genera including *Cutibacterium*, *Escherichia*, *Staphylococcus*, *Acinetobacter*, *Streptococcus*, and *Corynebacterium*, but not *Lactobacillus*, as a core endometrial microbiota [39]. The detection of *Bifidobacterium* in EF implies that the potential contamination of VS and endocervical secretions cannot be avoided in the process of trans-vagino-cervical EF sampling, even after thorough disinfection and cleansing. *Rhodanobacter* was detected in 15.0% of the EF microbiota in the CE group. The pathogenic role and significance of *Rhodanobacter* in CE remains unclear. To the best of our knowledge, there is only one case report that *Rhodanobacter* was detected in the endometrial microbiota of an infertile woman with recurrent reproductive failure [55,56]. 

Previous studies that obtained endometrial samples via the trans-peritoneo-myometrial route (e.g., laparotomy) demonstrated that the estimated bacterial load in the vaginal cavity is 100- to 10,000-fold more than those in the uterine cavity [27]. Consistent with our recent report [36], we optimized the amplification and sequencing conditions on this premise to simultaneously analyze various biomass samples such as EF and VS. Under these optimized protocols, however, the number of the OTU-assigned sequences in the VS microbiota of infertile women obtained via the trans-vagino-cervical route was yet 2.2-fold higher than those in the EF microbiota on average. Along with the inevitable contamination of the VS microbiota into the EF microbiota by trans-vagino-cervical EF sampling, such a difference in bacteria loads between VS and EF prompted us to investigate whether the VS microbiota can provide more information as a potential predictor for the presence or absence of CE than the EF microbiota.

Lactic acid is a major metabolic end product of carbohydrate fermentation produced via lactic acid bacteria and *Bifidobacterium* [32]. These bacterial genera exhibit a strong tolerance to acidity and benefit our health as an adjuvant to correct the intestinal dysbiosis induced by antibiotic administration or inflammatory bowel disease. In this study, we found that the detection rate and/or bacterial abundance of three lactic acid bacteria, *Streptococcus**, Enterococcus,* and *Atopobium,* are reduced in VS in infertile women with CE compared with the non-CE cohort, suggesting their contribution to the homeostasis and integrity of the vaginal environment.

Bacterial vaginosis is a pathologic condition recognized as uniform, homogeneous, gray-white to yellowish VS with pH 4.5 or greater and amine odor after adding potassium hydroxide, along with the presence of clue cells in the vaginal smear samples [57]. The local microbial profiling of bacterial vaginosis is characterized by a reduction in *Lactobacillus*, along with an increase in *Gardnerella*, *Atopobium*, *Prevotella*, *Peptostreptococcus*, *Mobiluncus*, *Sneathia*, *Leptotrichia*, *Mycoplasma*, and *Clostridiales*. We found that the detection rate and bacterial abundance of *Lactobacillus* in the VS and EF microbiota are similar between the CE and non-CE group. Additionally, the bacterial abundance of *Atopobium* and *Clostridiales* in the VS and/or EF microbiota was somewhat lower in CE than in the non-CE cohort. In accordance with previous studies, our results failed to obtain the evidence that bacterial vaginosis is associated with CE [11,58,59,60].

The bias of this research is (i) the small sample size; (ii) the exclusion of patients diagnosed by hysteroscopy only; (iii) the inclusion of three different types of cycles (natural, oocyte-pick up, and hormone replacement cycles); and (iv) the particular race/ethnicity studied (only Japanese population). These questions should be addressed in the future study.

In conclusion, this study disclosed that a reduction in three lactic acid bacteria (*Streptococcus*, *Enterococcus*, and *Atopobium*) and *Bifidobacterium*, but not *Lactobacillus*, in the VS microbiota is associated with the presence of CE in infertile women. Although the causality between the findings and CE remains unsolved, our results hold promise for the microbial diagnosis of CE, not by somewhat interventive EF collection, but by less invasive VS sampling.

## Figures and Tables

**Figure 1 diagnostics-12-00878-f001:**
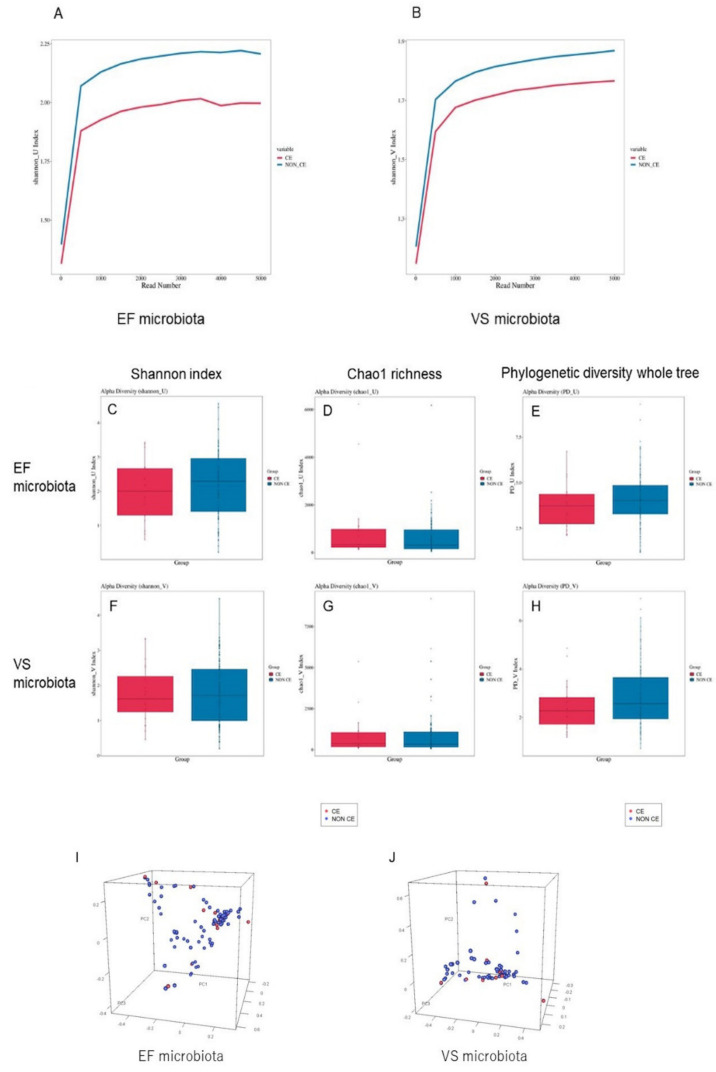
α- and β-Diversity values for comparison of bacterial communities in EF and VS in the CE and non-CE group. Rarefaction analysis of sequences per sample for Shannon index in EF (**A**) and VS (**B**) in CE (red lines) and non-CE (blue lines) cases. Comparison of Shannon index (**C**,**F**), Chao1 richness (**D**,**G**), and phylogenetic diversity whole tree (**E**,**H**) of EF (**C**–**E**) and VS (**F**–**H**) microbiota in the CE (red column) and non-CE (blue column). Each graph represents mean (column) and SE (bars). Principal coordinate analysis plotting for β-diversity of EF (**I**) and VS (**J**) microbiota in CE (red dots) and non-CE (blue dots) generated using UniFrac distance metrics.

**Figure 2 diagnostics-12-00878-f002:**
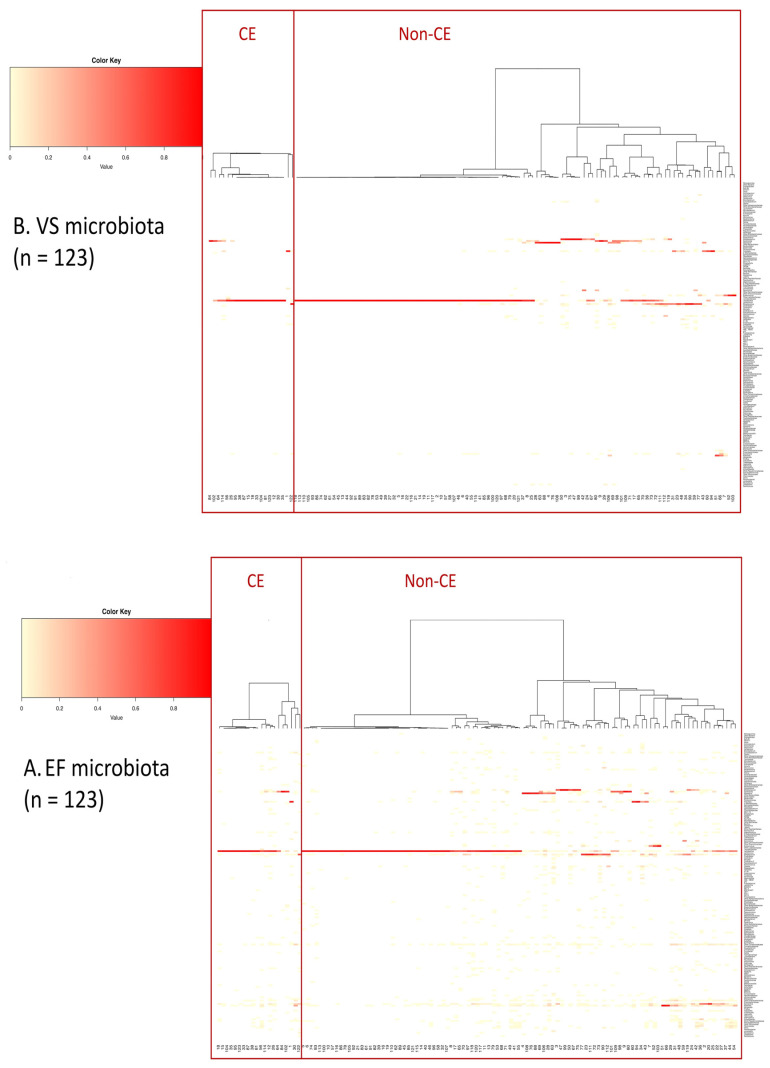
Heatmap diagrams representing dominant bacterial genera (with a total rate 0.1% or more) found in the EF (**A**) and VS (**B**) in the CE and non-CE groups. While the rows show bacterial genera, the columns represent the subject numbers.

**Table 1 diagnostics-12-00878-t001:** The rate of hysteroscopic CE findings proposed by International Working Group for Standardization of Chronic Endometritis Diagnosis [40] detected in patients with histopathologic CE.

Strawberry aspect	10% (2/20)
Focal hyperemia	20% (4/20)
Hemorrhagic spots	15% (3/20)
Micropolyposis	25% (5/20)
Stromal edema	30% (6/20)
Total	40% (8/20) ^a^

^a^ Totals are not 100 percent due to some patients with multiple endoscopic findings.

**Table 2 diagnostics-12-00878-t002:** Demographics of infertile patients in CE and non-CE group.

		CE Group (*n* = 20)	Non-CE Group (*n* = 103)	*p* Value ^c^
Age (years), mean ± SD		38.1 ± 3.7	38.4 ± 4.1	0.76
Body mass index (kg/m^2^), mean ± SD		22.1 ± 1.8	21.8 ± 2.3	0.58
Gravidity, median (range)		0 (0–3)	0 (0–4)	0.65
Parity, median (range)		0 (0–1)	0 (0–1)	0.84
Indication of IVF-ET ^a^				
	Male factor	4 (20.0%)	29 (28.2%)	0.59
	Polycystic ovarian syndrome	1 (5.0%)	16 (15.5%)	0.30
	Endometriosis	3 (15.0%)	25 (24.3%)	0.56
	Tubal factor	4 (20.0%)	10 (9.7%)	0.24
	Unexplained	8 (40.0%)	19 (18.4%)	0.042
	Diminished Ovarian Reserve	3 (15.0%)	35 (34.0%)	0.12
Controlled ovarian stimulation protocol history ^b^				
	Short gonadotropin-releasing hormone agonist cycle	12 (60.0%)	37 (35.9%)	0.0503
	Long gonadotropin-releasing hormone agonist cycle	0 (0.0%)	1 (0.1%)	1.00
	Flexible GnRH antagonist cycle	15 (80.0%)	82 (79.6%)	0.76
	Mild stimulation cycle	4 (20.0%)	33 (32.0%)	0.42
	Natural cycle	0 (0.0%)	4 (3.9%)	1.00
Past embryo transfer history, mean ± SD				
	Number of cycles	4.7 ± 3.6	4.4 ± 2.9	0.69
	Number of embryos transferred	5.9 ± 4.5	5.3 ± 3.8	0.53

^a^ Totals are not 100 percent due to some patients having more than one diagnosis; ^b^ Totals are not 100 percent due to some patients undergoing more than one controlled ovarian stimulation cycle; ^c^ Values of two-sided test(s).

**Table 3 diagnostics-12-00878-t003:** List of bacterial genera of which detection rate and/or bacterial abundance in EF and VS microbiota significantly differed between the CE and non-CE group.

	CE Group (*n* = 20)	Non-CE Group (*n* = 103)	Fisher’s Exact Test *p* Value	Wilcoxon Rank-Sum Test *p* Value	Adjusted Welch’s *t* Test *p* Value
EF microbiota					
*Rhodanobacter*	3 (15.0%)	2 (1.94%)	0.030	0.0062	0.12
*Atopobium*	2 (10.0%)	24 (23.3%)	0.24	0.13	0.0022
*Bifidobacterium*	2 (10.0%)	19 (18.5%)	0.52	0.30	0.0054
*Aeromonadaceae*	0 (0.0%)	11 (10.7%)	0.21	0.13	0.014
*Vibrio*	2 (10.0%)	19 (18.5%)	0.52	0.33	0.037
*Clostridiales*	0 (0.0%)	11 (10.7%)	0.69	0.41	0.048
VS microbiota					
*Streptococcus*	1 (5.0%)	40 (38.8%)	0.0033	0.0072	0.054
*Enterococcus*	0 (0.0%)	18 (17.5%)	0.042	0.045	0.67
*Atopobium*	0 (0.0%)	16 (15.5%)	0.072	0.061	0.0029
*Bifidobacterium*	1 (5.0%)	18 (17.5%)	0.31	0.15	0.0044
*Nesterenkonia*	0 (0.0%)	6 (5.8%)	0.59	0.27	0.019
*Anaerococcus*	0 (0.0%)	9 (8.7%)	0.35	0.17	0.024
*Staphylococcus*	0 (0.0%)	5 (4.9%)	0.59	0.32	0.033

## Data Availability

Data is contained within the article.

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
