# Peer review of "Differential Vaginal Microbiota Profiling in Lactic-Acid-Producing Bacteria between Infertile Women with and without Chronic Endometritis"

_diagnostics, 2022, doi:10.3390/diagnostics12040878_

Round 1
Reviewer 1 Report
Attached PDF with comments.
Author Response
Prof. Dr. Andreas Kjaer
Editor-in-Chief
Diagnostics
Dear Dr. Kjaer,
I (Yoshiyuki Sakuraba, PhD) and my co-authors (Suguru E Tanaka, PhD; Kotaro Kitaya, MD; and Tomomoto Ishikawa, MD) are resubmitting a revised manuscript entitled “Differential Vaginal Microbiota Profiling in Lactic Acid-producing Bacteria between Infertile Women with and without Chronic Endometritis” for publication in your esteemed journal Diagnostics.
We thank all the editors and peer-reviewers for their time and efforts they put in for this submission. We revised our manuscript incorporating the review comments, which are great help. Please find the Answers to Review Comments below in this letter.
About Specific comment 1 of Reviewer 1 "It should be included a link to see Figures 1 and 2 in an enlarged form since, due to space issues, it cannot be clearly read in the article", we need the judgement (for adoption) and assistance (if required) of the editorial office.
Thank you in advance.
Sincerely yours,
Yoshiyuki Sakuraba
Yoshiyuki Sakuraba, Ph.D.
Varinos Inc.
DiverCity Tokyo Office Tower, 12F, 1-1-20 Aomi, Koto-ku, Tokyo, Japan, 135-0064
TEL: +81 3 5422 6501
FAX: +81 3 5422 6501
E-mail: ysakuraba@varinos.com
Answers to Review Comments
Reviewer 1
Comments. Overall it is a well-designed and comprehensive study, since it compares histopathological and hysteroscopic findings with the use of next-generation sequencing techniques, thus minimizing interobserver variability and obtaining reliable results. In addition, vaginal and endometrial samples are taken from the same patients to detect cases of vaginal contamination of the endometrial sampling and compare both microbiota in each patient.
A. Thank you for your positive comments.
Suggestions. As a result of this study, it would be advisable to analyze the role of the depletion of other lactic acid producing bacteria in the genesis of chronic endometritis and its relationship with infertility.The data obtained in this study is insufficient to implement vaginal sampling in clinical practice as a way of inferring the endometrial microbiota without direct endometrial sampling. More studies are needed in this regard. In any case, in the conclusions it is appropriately indicated as "our results hold promise for microbial diagnosis of CE, not by somewhat interventive EF collection, but by less invasive VS sampling". The sampling of the patients was carried out mainly under the influence of hormonal treatments for assisted reproduction, which could have altered the basal microbiota of the patients. In future studies, it would be interesting to take samples in natural cycles in all patients. -Both groups of patients were homogeneous except in cases of infertility of unknown origin, which was higher in patients with chronic endometritis in a statistically significant way, so this fact should be taken into account in future studies so that it is not a confusion factor. -The results of the study are only applicable to the Japanese population since, as has been shown in other studies (25), there is significant variability in the vaginal microbiota of women of different races. The genus Streptococcus and Enterococcus was found in less abundance in the vaginal samples of some patients with chronic endometritis but not in most of them, so it can only be taken as a risk factor more associated with endometritis. It cannot yet be inferred that this disease exists in fact just with vaginal sampling without the direct taking of an endometrial sample.
A. Thank you for pointing out insufficiency in our study. Yes, we agree that more studies are needed to implement vaginal sampling in clinical practice as a way of inferring the endometrial microbiota. As you commented, all we can only say now is that our results hold promise for microbial diagnosis of CE, not by somewhat interventive EF collection, but by less invasive VS sampling". To refer to these bias and limitations of our studies, we add a paragraph and phrase in Discussion (line 396-402) as followings;
"The bias of this research is (i) small sample size, (ii) exclusion of patients diagnosed by hysteroscopy only (iii) inclusion of three different types of cycles (natural, oocyte-pick up, and hormone replacement cycles), and (iv) particular study race/ethnicity (only Japanese population). These questions should be addressed in the future study.
In conclusion, this study disclosed that reduction in Streptococcus and Enterococcus, but not Lactobacillus, in the VS microbiota is associated with the presence of CE in infertile women. Although the causality between our findings and CE remains unsolved, our results hold promise for microbial diagnosis of CE, not by somewhat-interventive EF collection, but by less invasive VS sampling".
Specific comments
1. It should be included a link to see Figures 1 and 2 in an enlarged form since, due to space issues, it cannot be clearly read in the article.
A. We are sorry for inconvenience but we have to consult the editorial office about this matter.
2. Reduce the number of self-citations, especially in articles by Kitaya K:
A. We deleted Reference no. 5, 7, and 56 (all by Kitaya K et al.,) as these can be covered by other references. We changed the reference numbers accordingly.
Reviewer 2 Report
I read with great interest the manuscript, which falls within the aim of this Journal. In my honest opinion, the topic is interesting enough to attract the readers’ attention. Nevertheless, authors should clarify some points and improve the discussion, as suggested below.
Authors should consider the following recommendations:
- Manuscript should be further revised in order to correct some typos and improve style.
- I suggest to add novel pieces of information about the diagnosis and management, as well as reproductive outcomes, in case of chronic endometritis (authors may refer to: PMID: 32802019; PMID: 31710184).
Author Response
Prof. Dr. Andreas Kjaer
Editor-in-Chief
Diagnostics
Dear Dr. Kjaer,
I (Yoshiyuki Sakuraba, PhD) and my co-authors (Suguru E Tanaka, PhD; Kotaro Kitaya, MD; and Tomomoto Ishikawa, MD) are resubmitting a revised manuscript entitled “Differential Vaginal Microbiota Profiling in Lactic Acid-producing Bacteria between Infertile Women with and without Chronic Endometritis” for publication in your esteemed journal Diagnostics.
We thank all the editors and peer-reviewers for their time and efforts they put in for this submission. We revised our manuscript incorporating the review comments, which are great help. Please find the Answers to Review Comments below in this letter. Thank you in advance.
Sincerely yours,
Yoshiyuki Sakuraba
Yoshiyuki Sakuraba, Ph.D.
Varinos Inc.,
DiverCity Tokyo Office Tower, 12F, 1-1-20 Aomi, Koto-ku, Tokyo, Japan, 135-0064
TEL: +81 3 5422 6501
FAX: +81 3 5422 6501
E-mail: ysakuraba@varinos.com
Answers to Review Comments
Reviewer 2
Comments and Suggestions. I read with great interest the manuscript, which falls within the aim of this Journal. In my honest opinion, the topic is interesting enough to attract the readers’ attention. Nevertheless, authors should clarify some points and improve the discussion, as suggested below. Authors should consider the following recommendations:
-Manuscript should be further revised in order to correct some typos and improve style.
-I suggest to add novel pieces of information about the diagnosis and management, as well as reproductive outcomes, in case of chronic endometritis (authors may refer to: PMID: 32802019)
A. Thank you for your positive comments. We corrected some typos in the text and quoted PMID: 32802019 as Reference no. 59. (Drizi A, et al., Impaired inflammatory state of the endometrium: a multifaceted approach to endometrial inflammation. Current insights and future directions. Prz Menopauzalny 2020; 19:90-100.)
Reviewer 3 Report
The authors reported that microbiota in vaginal secretions can be used as a potential prediction tool for chronic endometritis in infertile women since the reduction in the lactic acid-producing bacteria was observed. Although the microbiota are informative and helpful to predict and prevent the chronic endometritis, this study is too preliminary to be published.
Author Response
Prof. Dr. Andreas Kjaer
Editor-in-Chief
Diagnostics
Dear Dr. Kjaer,
I (Yoshiyuki Sakuraba, PhD) and my co-authors (Suguru E Tanaka, PhD; Kotaro Kitaya, MD; and Tomomoto Ishikawa, MD) are resubmitting a revised manuscript entitled “Differential Vaginal Microbiota Profiling in Lactic Acid-producing Bacteria between Infertile Women with and without Chronic Endometritis” for publication in your esteemed journal Diagnostics.
We thank all the editors and peer-reviewers for their time and efforts they put in for this submission. We revised our manuscript incorporating the review comments, which are great help. Please find the Answers to Review Comments below in this letter. Thank you in advance.
Sincerely yours,
Yoshiyuki Sakuraba
Yoshiyuki Sakuraba, Ph.D.
Varinos Inc.,
DiverCity Tokyo Office Tower, 12F, 1-1-20 Aomi, Koto-ku, Tokyo, Japan, 135-0064
TEL: +81 3 5422 6501
FAX: +81 3 5422 6501
E-mail: ysakuraba@varinos.com
Answers to Review Comments
Reviewer 3
Comments and Suggestions. The authors reported that microbiota in vaginal secretions can be used as a potential prediction tool for chronic endometritis in infertile women since the reduction in the lactic acid-producing bacteria was observed. Although the microbiota are informative and helpful to predict and prevent the chronic endometritis, this study is too preliminary to be published.
A. Thank you for your reasonable comments. We agree that more patients should be included along with reduction of bias and limitations in the future study. The comments are added in a paragraph (line 396-402).
"The bias of this research is (i) small sample size, (ii) exclusion of patients diagnosed by hysteroscopy only, (iii) inclusion of three different types of cycles (natural, oocyte-pick up, and hormone replacement cycles), and (iv) particular study race/ethnicity (only Japanese population). These questions should be addressed in the future study".
Round 2
Reviewer 3 Report
Comments and Suggestions. The authors reported that microbiota in vaginal secretions can be used as a potential prediction tool for chronic endometritis in infertile women since the reduction in the lactic acid-producing bacteria was observed. Although the microbiota are informative and helpful to predict and prevent the chronic endometritis, this study is too preliminary to be published.
>A. Thank you for your reasonable comments. We agree that more patients should be included along with reduction of bias and limitations in the future study. The comments are added in a paragraph (line 396-402).
"The bias of this research is (i) small sample size, (ii) exclusion of patients diagnosed by hysteroscopy only, (iii) inclusion of three different types of cycles (natural, oocyte-pick up, and hormone replacement cycles), and (iv) particular study race/ethnicity (only Japanese population). These questions should be addressed in the future study".
Responded comments:
My concern is mainly caused by Table 3. This table is the most convincing data in this study for readers for a possible diagnosis of chronic endometritis by different microbiota in vaginal secretions. However, the tested number of CE group (n =20) is problematic to compare with Non-CE group (n =103). However, I agree to conclude the top four listed bacteria in VS microbiota (Streptococcus, Enterococcus, Atopobium, Bifidobacterium) are significantly reduced compared to Non-CE group.
Table3:
The authors should double-check the number in this table. Enterococcus, Anaerococcus, Staphylococcus in CE group of the VS samples should be 0% detection rate (0/20, 0/20, 0/20 =0 %, not 10, 10, 5%).
Page 11, line 324-328:
“Meanwhile, Lactobacillus was detected in 84.5% (87/103) of the VS microbiota in the CE group, whereas it was detected in 90.0% (18/20) of the VS microbiota in the non-CE group.”
I think the data in this sentence is mixed up. CE group is n =20 and Non-CE group is n =103. I agree Lactobacillus species are abundant in both groups. Thus, the other lactic acid-producing bacteria are probably minor inhabitants in vagina in both groups.
Page 11, line 330-338:
“We focused on the combination of three lactic acid-producing bacteria Streptococcus, Enterococcus, and Bifidobacterium. The specificity for the diagnostic value of CE was 0.95 when the total abundance of the combination of these lactic acid-producing bacterial genera in the VS microbiota was set at 14.1% (p = 0.042 by Fisher’s exact test, sensitivity 0.26, accuracy 0.37). For prediction of CE, the minimal p value (p = 0.0023, specificity 0.90, sensitivity 0.47, accuracy 0.54) was obtained when the combined proportion of Streptococcus, Enterococcus, and Bifidobacterium in the VS microbiota was set at 0.1%.”
I understand the authors’ focus is lactic acid-producing bacteria, but if Atopobium is included for this prediction, is it going to improve the accuracy of the CE prediction? As the authors discussed “Additionally, the bacterial abundance of Atopobium and Clostridiales in the VS and/or EF microbiota was rather lower in CE than in the non-CE cohort,” Atopobium may contribute to the prediction.
Author Response
Answers to comments (by Reviewer 3)
Q. My concern is mainly caused by Table 3. This table is the most convincing data in this study for readers for a possible diagnosis of chronic endometritis by different microbiota in vaginal secretions. However, the tested number of CE group (n =20) is problematic to compare with Non-CE group (n =103). However, I agree to conclude the top four listed bacteria in VS microbiota (Streptococcus, Enterococcus, Atopobium, Bifidobacterium) are significantly reduced compared to Non-CE group.
A. Thank you for your comment.
Q. Table3: The authors should double-check the number in this table. Enterococcus, Anaerococcus, Staphylococcus in CE group of the VS samples should be 0% detection rate (0/20, 0/20, 0/20 =0 %, not 10, 10, 5%).
Page 11, line 324-328: “Meanwhile, Lactobacillus was detected in 84.5% (87/103) of the VS microbiota in the CE group, whereas it was detected in 90.0% (18/20) of the VS microbiota in the non-CE group.” I think the data in this sentence is mixed up. CE group is n =20 and Non-CE group is n =103.
A. Thank you very much for pointing out the misdescriptions.
We corrected Table 3 as follows: Enterococcus 0 (0.0%), Anaerococcus 0 (0.0%), Staphylococcus 0 (0.0%) in CE group of the VS samples.
We also corrected line 269-270 as follows: Lactobacillus was detected in 90.0% (18/20) of the EF microbiota in the CE group, whereas it was detected in 94.2% (97/103) of the EF microbiota in the non-CE group.
Q. Page 11, line 330-338: If Atopobium is included for this prediction, is it going to improve the accuracy of the CE prediction? As the authors discussed “Additionally, the bacterial abundance of Atopobium and Clostridiales in the VS and/or EF microbiota was rather lower in CE than in the non-CE cohort,” Atopobium may contribute to the prediction.
A. Thank you for your excellent suggestion.
We performed additional analysis using combination of “four” lactic acid-producing bacteria Streptococcus, Enterococcus, Bifidobacterium and “Atopobium” in the VS microbiota between infertile patients with and without CE. In result, the predictive values improved, as you pointed out.
These results were added in Line 304-312 as follows:
“We focused on the combination of four lactic acid-producing bacteria Streptococcus, Enterococcus, Bifidobacterium, and Atopobium, of which detection rate and/or bacterial abundance in the VS microbiota significantly differed between the CE group and non-CE group. The specificity for the diagnostic value of CE was 0.96 when the total abundance of the combination of these lactic acid-producing bacterial genera in the VS microbiota was set at 12.7% (p = 0.038 by Fisher’s exact test, sensitivity 0.29, accuracy 0.41). For prediction of CE, the minimal p value (p = 0.0021, specificity 0.91, sensitivity 0.47, accuracy 0.55) was obtained when the combined proportion of Streptococcus, Enterococcus, Bifidobacterium, and Atopobium in the VS microbiota was set at 0.1%”.
Discussion was also added as follows:
Line 381-383: “In this study, we found that the detection rate and/or bacterial abundance of three lactic acid bacteria Streptococcus, Enterococcus, and Atopobium are reduced in VS in infertile women with CE compared with non-CE cohort”, …
Line 401-403: “In conclusion, this study disclosed that reduction in three lactic acid bacteria (Streptococcus, Enterococcus, and Atopobium) and Bifidobacterium, but not Lactobacillus, in the VS microbiota is associated with the presence of CE in infertile women.
Accordingly, we rewrote Abstract section (Line 10-28) as follows:
“Purpose: Chronic endometritis (CE) is an infectious and in-flammatory disorder associated with infertility of unknown etiology, re-peated implantation failure, and recurrent pregnancy loss. In the current clinical practice, intrauterine interventions such as endometrial biop-sy/histopathologic examinations and/or hysteroscopy are required for the diagnosis of CE. In this study, we sought for the microbiota in the vaginal secretions (VS) as a potential prediction tool for CE in infertile women. Methods: Using next-generation sequencing analysis, we compared the VS and endometrial fluid (EF) microbiota in infertile women with (n =20) or without CE (n = 103). Results: The detection rate of Streptococcus and Enterococcus as well as the bacterial abundance of Atopobium and Bifidobacterium in the VS microbiota was significantly lower in the CE group than in the non-CE group. Meanwhile, the detection rate and bac-terial abundance of Lactobacillus in the EF and VS microbiota was at a similar level between the two groups. Conclusion: These findings suggest that VS microbiota in infertile women with CE is characterized by the reduction in Bifidobacterium and the lactic acid-producing bacteria other than Lactobacillus. Our results hold promise for prediction of CE, not by somewhat-interventional intrauterine procedures, but by less invasive VS sampling.”.
Round 3
Reviewer 3 Report
My comments are written in red in an attached Word file. My concern on Table 3 for the conclusion has been solved. The manuscript is now much better than the previous version.

Author Response
Answers to comments (by Reviewer 3)
A. Page 11, line 324-328 and Table 3, and Page 11, line 330-338:
The authors corrected properly. Glad to hear that including Atopobium improved the prediction. The authors improved the manuscript significantly, and now it is much better than the previous version. Overall, this manuscript suggests a potential diagnosis method for Chronic endometritis and this study will be helpful for the future clinical practice.
Q. Thank you again for your really great help to improve our study and acceptance of our text.
Q. Minor comment:
Figure 2 is impossible to read. I suggest the authors to improve this figure for readers.
A. We prepared the magnified version of Figure 2
Please check Figure 2A and 2B at the following links:
https://github.com/suguruetanaka/Tanakaetal2022_CE_LacticAcidBacteria/blob/b3cfb2dd206355a417bb1e6c4580178c0a04d8bd/images/Figure2A_EF_heatmap.jpg
https://github.com/suguruetanaka/Tanakaetal2022_CE_LacticAcidBacteria/blob/b3cfb2dd206355a417bb1e6c4580178c0a04d8bd/images/Figure2B_VS_heatmap.jpg
We are consulting with the editorial office about this matter. We expect they will go up as supplemental files.